# Salivary Gland Volume Changes and Dry Mouth Symptom Following Definitive Radiation Therapy in Oropharyngeal Cancer Patients—A Comparison of Two Different Approaches: Intensity-Modulated Radiation Therapy Versus Intensity-Modulated Radiation Therapy/Intensity-Modulated Proton Therapy Combination

**DOI:** 10.3390/cancers17030554

**Published:** 2025-02-06

**Authors:** Seung Gyu Park, Yong Chan Ahn, Dongryul Oh, Kyungmi Yang, Sang Gyu Ju, Jin Man Kim, Dongyeol Kwon, Euncheol Choi, Han Gyul Yoon

**Affiliations:** 1Department of Radiation Oncology, Keimyung University School of Medicine, Daegu 42601, Republic of Korea; psk818@dsmc.or.kr (S.G.P.); cec0510@dsmc.or.kr (E.C.); 2Department of Radiation Oncology, Samsung Medical Center, Sungkyunkwan University School of Medicine, Seoul 06351, Republic of Korea; dongryul.oh@samsung.com (D.O.); kyungmi.yang@samsung.com (K.Y.); sg.ju@samsung.com (S.G.J.); jmm.kim@samsung.com (J.M.K.); dy82.kwon@samsung.com (D.K.); 3Kim Jaechul Graduate School of Artificial Intelligence, Korea Advanced Institute of Science and Technology, Daejeon 34141, Republic of Korea; hangyulmd@kaist.ac.kr

**Keywords:** oropharyngeal cancer, radiation therapy, proton therapy, salivary gland, dry mouth

## Abstract

This study compared salivary gland volume changes and dry mouth symptom in 78 oropharyngeal cancer patients treated with IMRT alone or IMRT/IMPT combination. Ipsilateral parotid and submandibular glands showed significant volume reductions due to higher radiation doses, with the parotid stabilizing after an initial decline and the submandibular gland continuing to decline until 24 months. Contralateral salivary glands received lower doses, demonstrating recovery to baseline volumes by 48 months. Dry mouth symptom affected 60.3% of patients within 6 months, decreasing to 41.0% by 12 months, with no significant differences between groups. Volume reduction was greater in patients with dry mouth symptom. No significant differences in salivary gland volume changes or dry mouth symptom were observed between the two RT techniques, suggesting that the delivered dose to the salivary glands is the critical determinant rather than the RT modality.

## 1. Introduction

Worldwide, head and neck cancer (HNC) is the seventh most common cancer [1]. Over 60% of HNC patients require radiation therapy (RT), either alone or as a part of multimodality treatment regimens [2]. High-dose RT, though inevitably associated with the risk of surrounding normal tissue damage, has significantly contributed to improved clinical outcomes [3]. Approximately over 80% of HNC patients suffer from dry mouth symptom following high-dose RT, which results from altered salivary gland function and/or changes in saliva composition [4]. Several previous studies demonstrated the relationship between RT dose, salivary gland volumes, and dry mouth symptom, most of which were with relatively small sample sizes, short follow-up periods, and limited to photon-based RT techniques [5,6,7,8]. Although the theoretical advantages of proton beam therapy (PBT) have been widely proposed in treating patients with several HNC types, its actual application has been possible only in a limited number of institutes, and there still is a paucity of reports that focus on the impact of PBT on salivary gland volumes [9]. A few previous studies proposed theoretic benefits of PBT over photon-based RT techniques by reducing the risk of salivary gland damage in treating oropharyngeal cancer (OPC) patients, all of which, however, were not based on actual clinical data [10,11].

At our institution, the main RT technique in managing HNC patients used to be photon-based intensity-modulated radiation therapy (IMRT) by helical tomotherapy (HT) (TomoTherapy^®^, Madison, WI, USA) until 2015 (IMRT alone). Since late 2015, PBT has become clinically available at our institute. However, because of limited PBT resources when compared with clinical needs and the resultant long waiting time of around ≥6 weeks, PBT application to OPC patients throughout the whole RT course was not practically feasible. To avoid this undesirable long waiting period in treating eligible OPC patients, we designed a combination approach by starting RT with photon-based IMRT, which had a much shorter waiting time, and switching to intensity-modulated proton therapy (IMPT) during the later part of the RT course (IMRT/IMPT combination) in early 2016. This combination approach also aimed to take advantage of the physical profiles of both photon- and proton-based RT techniques while delivering a high enough dose to the targets effectively [12,13].

This study intended to carry out a comparative evaluation of salivary gland volume changes and their association with dry mouth symptom in OPC patients who underwent either IMRT alone or IMRT/IMPT as a combination approach.

## 2. Materials and Methods

After approval from the Institutional Review Board (IRB No. 2023-12-136), we retrospectively reviewed the medical records of OPC patients who underwent definitive RT from January 2008 to December 2020 at our institute. The inclusion criteria were (1) histologically confirmed squamous cell carcinoma of the oropharynx, (2) completion of curative-intent RT with or without concurrent systemic chemotherapy, and (3) inclusion of the ipsilateral neck lymphatics only within the RT target volume. Patients with a previous history of inflammatory salivary gland disorder, those known conditions or comorbidities requiring medications associated with medication-induced dry mouth or hyposalivation, or those who received non-concurrent (neoadjuvant or adjuvant) systemic therapy were excluded. A total of 78 patients formed the basis of the current study. We applied the 7th edition of the American Joint Committee on Cancer staging system, which better reflects the anatomic disease extent than the 8th edition. In the 7th edition, T, N, and overall stages were assigned based on the anatomic disease extents regardless of the human papillomavirus (HPV) relatedness. In the 8th edition, however, N and overall stages were assigned differently according to the HPV relatedness, though with the same anatomic disease extents, which intended to reflect the different prognostic profiles in HPV (−) and HPV (+) patients [14].

In all patients, RT was commenced with the concepts of simultaneous integrated boost (SIB) and adaptive re-plan, as previously described [12,13]. All patients underwent contrast-enhanced computed tomography (CT) scans twice with an individually custom-made tongue displacement device and thermoplastic mask fitting for RT plan simulation: the first one for the initial plan and the second one for the adaptive re-plan [15]. On each set of simulation CT scans, the gross tumor volumes (GTVs) of the primary tumor and metastatic lymph nodes were delineated based on all available clinical findings and diagnostic images. The initial CTV aimed to encompass the regions immediately adjacent to the GTV of the primary tumor and metastatic lymph nodes plus 1.5~2 additional lymphatic levels. The adaptive CTV was reduced to eliminate the overtly uninvolved lymphatic levels, which were covered in the initial CTV [13,16]. Both the GTV and CTV were adjusted considering the surrounding normal anatomical barriers. As a result, two sets of CTVs were generated: a high-risk clinical target volume (HR-CTV) that was consistently irradiated throughout entire RT course and a low-risk clinical target volume (LR-CTV) that was excluded during the adaptive re-plan. Two typical dose schemes used in the current study are summarized in Table 1. To accommodate the relative resource limitation at our institute around 2018, we reduced the total number of fractions from 30 to 28. The early dose schedule was applied to 50 patients (64.1%), while the modified schedule was to 28 patients (35.9%).

In all patients, the initial part of RT was delivered by IMRT. At the time of the adaptive re-plan, two sets of rival plans were generated for an objective comparison before the actual assignment of RT techniques: one was the IMRT plan and the other was the IMPT plan (RayStation^®^, RaySearch Laboratories AB, Stockholm, Sweden). For the IMRT plans, the planning target volumes (PTVs) for three target volumes were generated by adding 3 mm concentric margins. The plan conditions included a field width of 2.5 cm, modulation factor of 2.0, and pitch of 0.287. Dynamic jaw mode (TomoEDG-ETM^®^, Accuray, CA, USA) was employed to improve the longitudinal dose conformity [17]. For the IMPT plans, the relative biological effectiveness was considered as 1.1, and single-field optimization based on active line scanning (Sumitomo Heavy Industries^®^, Ltd., Tokyo, Japan), with the range shifter (4 cm water equivalent thickness), was employed. The beam arrangements for the IMPT plan typically involved two beams (one posterior port and one ipsilateral anterior oblique port) considering the configuration between the targets and organs-at-risk. The parameters for robust optimization for IMPT included 3-dimensional setup errors of ±3 mm and range uncertainties of ±3.5% of the nominal range to the GTV and CTV, which considered a total of 21 different scenarios. The radiation oncologist in charge determined whether each patient would receive IMRT or IMPT as the adaptive RT technique based on the dosimetric properties of comparative rival plans in addition to the availability of equipment.

Sixty-seven patients (85.9%) received concurrent systemic therapy with RT, while eleven (14.1%) underwent RT alone (Table 2). Cisplatin-based chemotherapy was applied in most patients: tri-weekly cisplatin (100 mg/m^2^, 2 cycles) in 63 patients (80.8%), and weekly cisplatin (20 mg/m^2^) + docetaxel (20 mg/m^2^) in 2 (2.6%). In two patients (2.6%), who were not fit for cytotoxic chemotherapy due to old age and medical comorbidities, cetuximab (400 mg/m^2^ as the initial loading dose and 250 mg/m^2^ thereafter) was used alternatively.

During the RT course, all patients were interviewed weekly to evaluate the acute toxicity profiles. The first response evaluation was performed with contrast-enhanced neck CT scans taken in 1 month of RT completion, and the second response evaluation was performed with fluorodeoxyglucose positron emission tomography/CT scans taken in 3~4 months thereafter. Subsequent regular follow-up evaluations, mainly with neck CT scans, were scheduled at 3~4-month intervals during the first two years and 6~12-month intervals thereafter. Dry mouth was evaluated based on clinical assessments documented in the medical records. The severity was graded according to the Common Terminology Criteria for Adverse Events version 5.0, which classifies dry mouth as follows: grade 1 (mild symptom, without dietary alteration), grade 2 (moderate symptom, with oral intake alterations), and grade 3 (inability to adequately aliment orally) [18].

All CT scans, taken for the RT plans and for follow-up evaluations at various time points, were transferred into the Eclipse Treatment Planning System (Varian Medical System^®^, Palo Alto, CA, USA). This study focused on the parotid gland (PG) and submandibular gland (SMG), which can be reliably visualized on routine CT scans. The sublingual glands were not included due to their relatively small size and the difficulties in consistent delineation on CT scans. The ipsilateral and contralateral PG and SMG were contoured and their volumes were measured on all CT scans for serial comparison. To ensure consistency and to minimize the inter-observer variability, all contouring tasks were carried out by a single radiation oncologist (SGP). The volumes of the salivary glands and dosimetric parameters, including the mean dose (D_mean_), V_10Gy_, V_20Gy_, V_30Gy_, V_40Gy_, V_50Gy_, and V_60Gy_ (volume covered by 10 Gy, 20 Gy, 30 Gy, 40 Gy, 50 Gy, and 60 Gy isodose surface of the prescription dose, respectively) to each salivary gland, were measured using the planning system. To compensate for the individual volume differences among patients, the volume ratio (VR: post-RT volume/initial volume) of each salivary gland was calculated.

The chi-square test or Fisher’s exact test for the categorical variables and the independent *t*-test for the continuous variables were used for the comparison of the patients’ characteristics between groups. Treatment outcomes of overall survival (OS) and disease-free survival (DFS) were compared using the log-rank test. Differences in the dosimetric parameters were compared using independent *t*-tests. The changes in the salivary gland volumes over time were assessed using paired *t*-tests. To compare the changes in salivary gland volumes between different treatment modalities or dry mouth symptom, repeated-measure ANOVA was used. A *p*-value < 0.05 was considered statistically significant. Statistical analysis was performed using SPSS ver. 28.0 (SPSS Inc.^®^, Chicago, IL, USA).

## 3. Results

### 3.1. Patients’ Characteristics

Among 78 patients, 39 (50.0%) received IMRT alone, while 39 (50.0%) received the IMRT/IMPT combination. The median age of all patients was 60 years (range, 39~89 years), and the majority were male (67, 85.9%). The demographic and clinical characteristics were well balanced between groups (Table 2). The majority of patients had stage-IV disease (55, 70.5%), due to the high incidence of N2- or N3-stage disease, without statistical difference between groups. There was no significant difference in the GTVs between groups (20.9 ± 10.5 cc vs. 24.6 ± 33.3 cc, *p* = 0.511) or the proportions of patients who received CCRT (89.7% vs. 82.1%, *p* = 0.329).

### 3.2. Oncologic Outcomes

The median follow-up time of all patients was 62 (6~160) months: 71 (17~160) months and 53 (6~84) months for IMRT alone and IMRT/IMPT combination groups, respectively. The 5-year rates of OS and DFS in all patients were 93.8% and 88.7%, with no differences between groups (91.9% vs. 93.8%, *p* = 0.403; and 92.0% vs. 82.9%, *p* = 0.441).

### 3.3. Dosimetric Parameters

The dosimetric parameters of the salivary glands are summarized in Table 3. The Dmean to the ipsilateral PG was significantly lower in the IMRT-alone group (24.8 ± 7.4 Gy vs. 30.3 ± 10.4 Gy, *p* = 0.011). Also, V_20 Gy_ and V_30 Gy_ were significantly lower in the IMRT-alone group (48.5 ± 14.6% vs. 59.7 ± 18.1%, *p* = 0.008, and 34.9 ± 13.5% vs. 45.3 ± 21.2%, *p* = 0.024, respectively). And the doses delivered to the contralateral PG and SMG were significantly higher in IMRT-alone group (8.2 ± 5.4 Gy vs. 3.5 ± 0.9 Gy, *p* < 0.001, and 11.3 ± 5.2 Gy vs. 6.7 ± 3.0 Gy, *p* < 0.001, respectively). V_10 Gy_ and V_20 Gy_ to the contralateral PG and V_10 Gy_ to the contralateral SMG were also significantly higher in the IMRT-alone group (20.8 ± 22.8% vs. 8.1 ± 7.9%, *p* = 0.003, 2.1 ± 5.0% vs. 0.0 ± 0.1%, *p* = 0.015, and 51.8 ± 32.6% vs. 33.8 ± 13.3%, *p* = 0.004, respectively).

When comparing the Dmean by adaptive plans separately, which could directly compare the IMRT and IMPT plans, IMRT plans were to deliver significantly lower Dmean to the ipsilateral PG (9.7 ± 3.1 Gy vs. 13.0 ± 4.4 Gy, *p* < 0.001), while delivering a significantly higher Dmean to the contralateral PG (2.7 ± 0.1 Gy vs. 0.0 ± 0.0 Gy, *p* < 0.001) and to the contralateral SMG (4.1 ± 1.8 Gy vs. 0.2 ± 0.6 Gy, *p* < 0.001), respectively.

### 3.4. Salvary Gland Volume Changes

The VR changes of salivary glands over time are shown in Figure 1. There was an initial sharp decline in the VR of the ipsilateral PG shortly after RT started, reaching 0.74 in 1 month of RT ending (*p* < 0.001). Following this initial sharp decline, the VR stabilized between 0.77 and 0.81 throughout the follow-up period. In contrast, there was a continuous decline in the VR of the ipsilateral SMG until 24 months after RT ended (*p* < 0.001), which decreased continuously and reached approximately 0.47 by 48 months. The VRs of the contralateral PG and SMG initially decreased (up to 0.86 of the initial volume by 1 month after RT end) and subsequently showed gradual recovery, reaching almost the initial volume (1.00 and 0.97 for the contralateral PG and SMG, respectively) in 48 months of RT ending. The changes in the VRs over time according to the treatment groups are shown in Figure 2, which indicate no significant differences over time between the two groups.

### 3.5. Changes in Dry Mouth Symptom

In 6 months after RT ending, 47 patients (60.3%) experienced grade-1–2 dry mouth: grade 1 in 45 (57.7%) and grade 2 in 2 (2.6%). The prevalence of grade-1–2 dry mouth slightly decreased after 12 months (32, 41.0%) and remained stable after 24 months (29, 37.2%). Both groups showed a relatively high prevalence of dry mouth symptom immediately after RT ending (71.8% in IMRT alone and 48.8% in the combination group, *p* = 0.110), which slightly decreased after 12 months (35.9% and 46.2%, *p* = 0.441) and then remained stable after 24 months (33.3% and 41.1%, *p* = 0.515).

Figure 3 illustrates the salivary gland VR change according to the presence of grade-1–2 dry mouth. The volume reductions over time in the ipsilateral salivary glands were greater in patients with dry mouth symptom, which suggested a correlation between salivary gland volume reduction and dry mouth symptom. However, the changes in the contralateral salivary gland volumes were not significantly associated with dry mouth symptom.

## 4. Discussion

Salivary gland damage occurs inevitably following high-dose RT to the head and neck regions, typically leading to symptomatic dry mouth. Even with the modern RT techniques, the majority of patients still experience some degree of dry mouth [3]. This annoying symptom is frequently associated with difficulties in mastication and swallowing and impaired taste and enhances the risk of dental problems. These symptoms usually remain unrecovered, are difficult to manage, and subsequently decrease the patients’ quality of life [19]. Both the PG and SMG are frequently located close to the target volumes and are prone to high-dose exposure, which inevitably can lead to various degrees of dry mouth symptom. Although the detailed mechanism of radiation-induced dry mouth is not fully known, high radiation doses to the salivary glands can cause atrophy and loss of acinar cells and granules, leading to reduced saliva production and morphological changes [20].

It is well known that the salivary gland volumes, dry mouth symptom, and RT dose are closely related to each other [6,7,21,22,23,24,25]. Teshima et al. reported that the decrease in PG volume was closely related with the reduction in saliva production in oral cancer patients following high-dose RT [6]. They found that a significant volume reduction in the PG after 30 Gy irradiation corresponded with a significant decrease in saliva production. Similarly, the SMG volumes also showed substantial reduction, around 38% of the pre-treatment volume, mainly during the first several months after RT [7]. The studies investigating the relationship between RT dose and salivary gland function reported that a Dmean <25~30 Gy to the PG could allow complete recovery of salivary flow rate [26,27]. Conversely, other studies revealed that a Dmean of 48 Gy to the PG resulted in a consistent decline in salivary function with moderate recovery over time, even after conformal RT techniques [28]. The relationship between the dose and salivary function recovery was investigated by Hey et al. [29], who evaluated the recovery potential of the PG in 117 patients and found that complete salivary flow rate recovery was expected if a Dmean of <26 Gy in at least one PG was achieved. Steenbakkers et al. recently conducted a double-blind randomized trial to test the impact of PG stem-cell-sparing RT [30]. The Dmean to the PG was 27 Gy in both arms, which was low enough, using recently improved RT techniques, and this study failed to demonstrate the expected advantage of the stem-cell-sparing effort, which signified the importance of the Dmean.

In the current study, we observed how the salivary gland volumes changed over time following RT. As our study included patients who received ipsilateral neck irradiation, the ipsilateral PG and SMG were typically located close to the target volumes and frequently overlapped with them, while the contralateral glands were not. The ipsilateral PG, which was exposed to moderate radiation doses (Dmean of 27.6 ± 9.4 Gy), showed a volume decline within one month and then plateaued. The ipsilateral SMG, which was exposed to higher radiation doses (Dmean of 56.0 ± 5.2 Gy), showed continuous volume decline without recovery. In contrast, the contralateral PG and SMG that received relatively lower radiation doses (Dmean of 5.7 ± 4.5 Gy and 8.9 ± 4.8 Gy, respectively) showed minimal changes within one month, which fully recovered later regardless of the RT techniques. These findings were consistent with those observed in previous studies.

We compared the effects of two different treatment approaches: IMRT alone versus IMRT/IMPT combination. Based on several advantages and disadvantages of PBT, different profiles of salivary gland volume changes over time and dry mouth symptom were theoretically expected, which, however, were negated through our study. The current study results are summarized as follows. First, although the dosimetric parameters for the two RT techniques were different, there were no significant differences in salivary gland volume changes. The Dmean of the ipsilateral PG showed a significant difference (24.8 ± 7.4 Gy vs. 30.3 ± 10.4 Gy, *p* = 0.011); however, no significant difference in volume change was observed. Similarly, though the Dmean of the contralateral PG and SMG was significantly reduced by the IMRT/IMPT combination (8.2 ± 5.4 Gy vs. 3.5 ± 0.9 Gy, *p* < 0.001, and 11.3 ± 5.2 Gy vs. 6.7 ± 3.0 Gy, *p* < 0.001, respectively), similar volume preservation was observed in both groups. In brief, although there were some numerical differences, these were not significant enough to cause the differences in volume change and degree of dry mouth symptom. This might be partly explained by the advancement of recent IMRT techniques, which could have played an important role in sparing the normal tissues successfully. Its precision in targeting the tumor while minimizing radiation exposure to the surrounding healthy tissues has been well documented [31]. As a result, the recommended dose-volume constraints of the normal tissues were achievable by both RT techniques in the current study, which resulted in dry mouth symptom that was comparable to the previous study reports [4,32,33].

Second, the consistent contouring and the implementation of SIB and adaptive re-plan policies in this study may have contributed to effective sparing of the salivary glands. The thorough consideration of anatomical barriers and exclusion of the overtly uninvolved level-Ib lymphatics from the CTV have been longstanding policies at our institute. This approach must have helped in reducing the dose delivered to the SMG, which resulted in a very low incidence of grade-2 or higher dry mouth symptoms. Moreover, the implementation of SIB and adaptive re-plan policies could have contributed to less severe salivary gland damage by accommodating the changes in the target and body contour in relation to the setup uncertainties during the RT course [5,34].

Third, the potential risk of PBT delivering higher doses to the nearby organs, due to their physical properties, is noteworthy. When compared to photon-based RT techniques, PBT can offer superior dose distributions in the nearby regions by virtue of rapid dose fall-off. However, this physical advantage usually becomes less dramatic, even meaningless, in the regions that are very close to or abutting the targets [35,36]. This is because of broader lateral dose fall-off, mimicking the wide penumbra width in photon-based RT techniques, which typically increases with depth, especially at the Bragg peak. This phenomenon is frequently associated with the use of range shifters and the presence of air gaps, which are very commonly encountered when treating most HNC patients. The IMPT port arrangement frequently makes it difficult to avoid passing through the ipsilateral salivary glands, typically in conditions when patients have metallic artifacts, which are much more sensitive to dose calculation in IMPT than in IMRT. As a result of limitations in equipment and tissue heterogeneity, IMPT often results in higher dose delivery to the nearby normal tissues than IMRT [37,38]. Moreover, if the entire treatment course had been conducted with IMPT from the beginning, the ipsilateral salivary gland damage would likely have been more significant, while the contralateral glands would have been better preserved [39]. This might have led to different patterns in both volume changes and dry mouth symptoms. With improved contralateral gland sparing, one could speculate whether compensatory hyperfunction might occur in these glands, potentially mitigating the severity of dry mouth [30].

Furthermore, while IMRT using HT was based on full arc rotation, IMPT typically used two beam ports. Because oral mucositis and oral pain are regarded as more annoying acute side effects than dry mouth, the beam ports were frequently arranged to avoid passing through the oral cavity but rather through the ipsilateral salivary glands to alleviate severe oral mucositis [12,13]. Incorporating multiple beam ports or arc rotation techniques into IMPT could be advantageous. However, the inevitable increase in treatment time remains a practical challenge that must be addressed.

Considering all these factors, the most critical factor in salivary gland volume changes and dry mouth symptom seems to be the delivered dose to the salivary glands, but not the RT technique used.

The findings of this study provide insights into the relationship between radiation dose, salivary gland volume changes, and dry mouth symptom, which are critical for optimizing radiotherapy strategies in OPC patients. Clinically, these results emphasize the importance of dose-volume constraints for salivary glands to mitigate treatment-related toxicities, even when using advanced modalities such as IMRT or IMPT. For patients with higher risks of developing dry mouth symptom, such as those with pre-existing conditions affecting salivary gland function, our findings suggest the potential benefit of tailored treatment plans that prioritize sparing both ipsilateral and contralateral salivary glands. Furthermore, the study highlights the need for robust dosimetric planning and adaptive strategies to address inter-patient variability and gland response to radiation. Future studies with larger cohorts and longer follow-up are necessary to confirm these findings and explore the potential of novel techniques, such as advanced multi-beam proton therapy, to further reduce treatment-related toxicities while maintaining oncologic efficacy.

This study has several limitations. First, the retrospective study design might have posed the risk of over- or underestimating the subjective symptom of dry mouth. Dry mouth is a symptom perceived subjectively by the patient and does not necessarily reflect the objective change in the salivary gland function. The absence of grade-3 dry mouth might have been because we relied on the physician-reported toxicity data only instead of the patient-reported outcome or objective quantitative measurement of salivary flow. Second, since the salivary glands are known to be radiosensitive, a significant proportion of salivary gland damage could have already happened during the initial IMRT, which was shown by the reduced volume measured at the time of the adaptive re-plan. As all patients received initial IMRT, the current comparisons that mainly focus on the adaptive re-plan RT techniques may have had a less significant impact than expected. Third, along with the OPC treatment guidelines, most patients received systemic therapy concurrently with RT, which also could have affected the salivary gland volume and function changes and subsequent dry mouth symptom degree. Fourth, as previously mentioned, most treatment plans adhered to the recommended dose-volume constraints, resulting in minimal differences in dose distribution, which in turn might have led to a lack of significant differences in the outcomes. Fifth, all contouring and volume measurements were performed by a single radiation oncologist (SGP). While this ensured consistency, it does not eliminate the potential for intra-observer variability. Ideally, repeated measurements or multiple independent observers should be used to enhance the reliability of subjective evaluations. Future studies should incorporate methods such as blinded multi-observer validation or automated segmentation tools to further improve measurement reliability. Another limitation of this study may be the imbalance in the HPV status between two treatment groups. While the HPV status is a well-established prognostic factor in OPC, its direct impact on the salivary gland volume change and the severity of dry mouth symptom remains unclear. It may be worthwhile to explore this issue in future studies.

This study, however, has a few strengths. This study seems to be the first one that focused on the salivary gland volume changes and associated dry mouth symptoms based on two different RT techniques in real-world situations. Additionally, the salivary gland volume assessment was extended up to 48 months, allowing a comprehensive assessment of long-term salivary gland function.

## 5. Conclusions

This study compared salivary gland volume changes and dry mouth symptoms depending on the RT techniques (IMRT alone versus IMRT/IMPT combination) in treating OPC patients with ipsilateral neck irradiation. The ipsilateral salivary glands, which received moderate to high radiation doses, showed notable volume reductions, while the contralateral glands, which received low radiation doses, exhibited minimal volume changes. Both groups demonstrated no significant difference in salivary gland volume change and dry mouth symptom. These findings indicate that the critical factor in salivary gland volume change and dry mouth symptom was the delivered dose to the salivary glands, but not the RT technique used. Further research may be needed to further understand the potential benefits and limitations of these RT techniques.

## Figures and Tables

**Figure 1 cancers-17-00554-f001:**
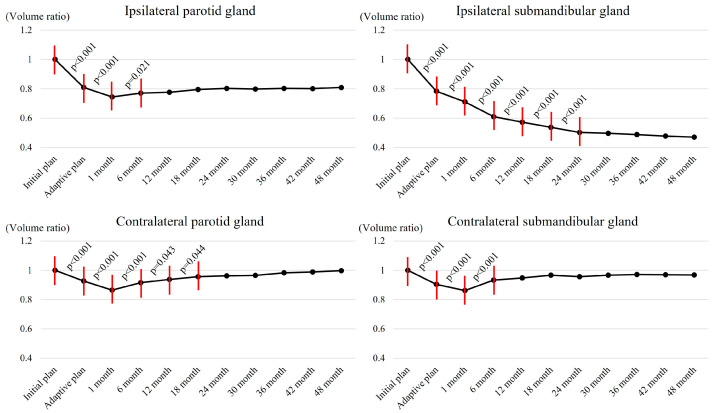
Changes in salivary gland volume ratio over time (post-RT volume/initial volume). Note: All months indicate the number of months after the end of treatment. Statistically significant intervals are noted with red lines.

**Figure 2 cancers-17-00554-f002:**
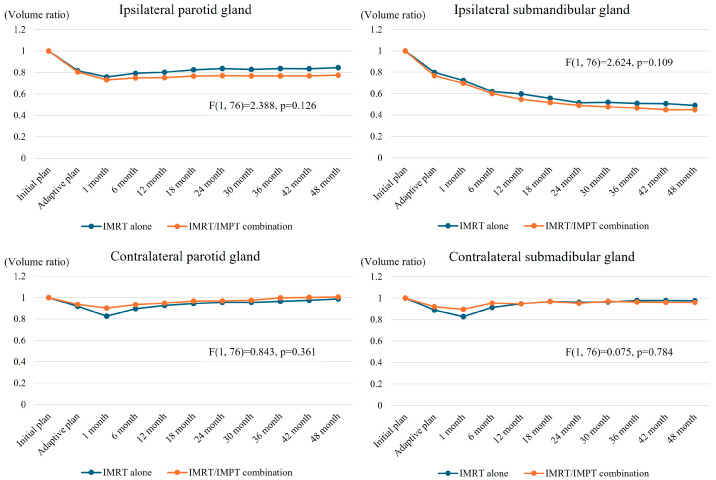
Changes in salivary gland volume ratio over time (post-RT volume/initial volume) according to the treatment groups. Note: All months indicate the number of months after the end of treatment.

**Figure 3 cancers-17-00554-f003:**
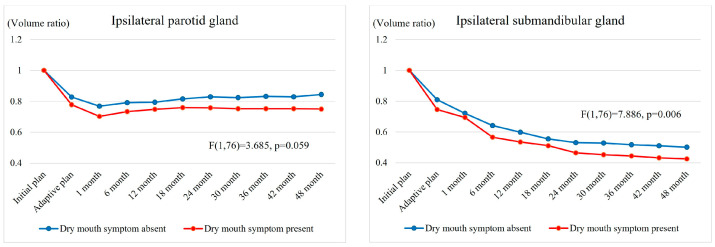
Changes in salivary gland volume ratio over time (post-RT volume/initial volume) according to the presence of dry mouth symptom. Note: All months indicate the number of months after the end of treatment.

**Table 1 cancers-17-00554-t001:** Dose schemes used through simultaneous integrated boost in adaptive re-plan policy.

	Early Scheme Until 2018 28 Patients (35.9%)			Later Scheme Since 2018 50 Patients (64.1%)		
	Initial Plan	Adaptive Re-Plan	Total Dose (EQD2)	Initial Plan	Adaptive Re-Plan	Total Dose (EQD2)
GTV	2.2 Gy × 18 Fxs	2.4 Gy × 12 Fxs	68.4 Gy (70.0 Gy)	2.4 Gy × 16 Fxs	2.4 Gy × 12 Fxs	67.2 Gy (69.4 Gy)
HR-CTV	2.0 Gy × 18 Fxs	2.0 Gy × 12 Fxs	60.0 Gy (60.0 Gy)	2.0 Gy × 16 Fxs	2.0 Gy × 12 Fxs	56.0 Gy (56.0 Gy)
LR-CTV	2.0 Gy × 18 Fxs	--	36.0 Gy (36.0 Gy)	2.0 Gy × 16 Fxs	--	32.0 Gy (32.0 Gy)

CTV, clinical target volume; EQD2, equivalent dose in 2 Gy fractions, calculated using the α/β ratio of 10; Fxs, fractions; GTV, gross tumor volume; HR, high risk; LR, low risk.

**Table 2 cancers-17-00554-t002:** Baseline demographic and clinical characteristics.

		All Patients	IMRT Alone Group	IMRT/IMPT Combination Group	
		(n = 78)	(n = 39)	(n = 39)	*p*-Value
Age (years)					
	Mean ± SD	59.4 ± 10.1	58.4 ± 10.4	60.4 ± 9.9	0.202
	Median (range, years)	60 (39~89)	60 (40~85)	59 (39~89)	
Gender					
	Male	67 (85.9%)	35 (89.7%)	32 (82.1%)	0.329
	Female	11 (14.1%)	4 (10.3%)	7 (17.9%)	
Smoking history					
	>10 pack-years	51 (65.4%)	29 (74.4%)	22 (56.4%)	0.096
	≤10 pack-years	27 (34.6%)	10 (25.6%)	17 (43.6%)	
Primary tumor size (cm)					
	Mean ± SD	2.2 ± 0.9	2.2 ± 0.9	2.1 ± 0.9	0.300
Largest LN size (cm)					
	Mean ± SD	2.8 ± 1.2	2.9 ± 1.2	2.6 ± 1.2	0.244
HPV status					
	Positive	60 (76.9%)	25 (64.1%)	35 (89.7%)	0.007
	Negative	3 (3.8%)	1 (2.6%)	2 (5.1%)	
	Unknown	15 (19.2%)	13 (33.3%)	2 (5.1%)	
AJCC 7th T stage					
	cT1	36 (46.2%)	18 (46.2%)	18 (46.2%)	1.000
	cT2	42 (53.8%)	21 (53.8%)	21 (53.8%)	
AJCC 7th N stage					
	cN0	7 (9.0%)	4 (10.3%)	3 (7.7%)	0.184
	cN1	16 (20.5%)	6 (15.4%)	10 (25.6%)	
	cN2	52 (66.7%)	29 (74.4%)	23 (59.0%)	
	cN3	3 (3.8%)	0 (0.0%)	3 (7.7%)	
AJCC 7th stage					
	Stage I	2 (2.6%)	2 (5.1%)	0 (0.0%)	0.339
	Stage II	5 (6.4%)	2 (5.1%)	3 (7.7%)	
	Stage III	16 (20.5%)	6 (15.4%)	10 (25.6%)	
	Stage IV	55 (70.5%)	29 (74.4%)	26 (66.7%)	
GTV (cc)					
	Mean ± SD	22.8 ± 24.6	20.9 ± 10.5	24.6 ± 33.3	0.511
Treatment					
	CCRT	67 (85.9%)	35 (89.7%)	32 (82.1%)	0.329
	RT alone	11 (14.1%)	4 (10.3%)	7 (17.9%)	

AJCC, American Joint Committee on Cancer; CCRT, concurrent chemoradiotherapy; GTV, gross tumor volume; HPV, human papillomavirus; IMRT, intensity-modulated radiation therapy; IMPT, intensity-modulated proton therapy; LN, lymph node; RT, radiation therapy; SD, standard deviation.

**Table 3 cancers-17-00554-t003:** Comparison of dosimetric parameters for salivary glands.

	IMRT Alone	IMRT/IMPT Combination	*p*-Value
Ipsilateral parotid gland			
Mean dose (Gy)	24.8 ± 7.4	30.3 ± 10.4	0.011
V_10Gy_ (%)	81.3 ± 47.9	77.0 ± 14.2	0.614
V_20Gy_ (%)	48.5 ± 14.6	59.7 ± 18.1	0.008
V_30Gy_ (%)	34.9 ± 13.5	45.3 ± 21.2	0.024
V_40Gy_ (%)	23.9 ± 11.8	31.8 ± 21.6	0.079
V_50Gy_ (%)	15.6 ± 8.8	16.9 ± 10.9	0.600
V_60Gy_ (%)	8.6 ± 5.3	10.4 ± 9.5	0.345
Ipsilateral submandibular gland			
Mean dose (Gy)	56.3 ± 4.8	55.8 ± 5.5	0.689
V_10Gy_ (%)	100.0 ± 0.0	98.4 ± 8.3	0.294
V_20Gy_ (%)	99.6 ± 1.2	97.5 ± 8.4	0.178
V_30Gy_ (%)	96.5 ± 5.1	94.5 ± 9.4	0.300
V_40Gy_ (%)	85.5 ± 12.0	83.4 ± 16.1	0.559
V_50Gy_ (%)	61.3 ± 15.8	54.9 ± 17.8	0.135
V_60Gy_ (%)	49.6 ± 66.7	34.4 ± 10.9	0.183
Contralateral parotid gland			
Mean dose (Gy)	8.2 ± 5.4	3.5 ± 0.9	<0.001
V_10Gy_ (%)	20.8 ± 22.8	8.1 ± 7.9	0.003
V_20Gy_ (%)	2.1 ± 5.0	0.0 ± 0.1	0.015
V_30Gy_ (%)	0	0	NA
V_40Gy_ (%)	0	0	NA
V_50Gy_ (%)	0	0	NA
V_60Gy_ (%)	0	0	NA
Contralateral submandibular gland			
Mean dose (Gy)	11.3 ± 5.2	6.7 ± 3.0	<0.001
V_10Gy_ (%)	51.8 ± 32.6	33.8 ± 13.3	0.004
V_20Gy_ (%)	11.2 ± 16.0	4.5 ± 11.8	0.058
V_30Gy_ (%)	0.7 ± 2.4	1.5 ± 5.4	0.466
V_40Gy_ (%)	0.2 ± 1.3	0.3 ± 1.2	0.856
V_50Gy_ (%)	0.1 ± 0.6	0.0 ± 0.1	0.335
V_60Gy_ (%)	0.0 ± 0.2	0.0 ± 0.0	0.277

Note: Values are presented as mean ± standard deviation. V_10Gy_, V_20Gy_, V_30Gy_, V_40Gy_, V_50Gy_, and V_60Gy_ are volume covered by 10 Gy, 20 Gy, 30 Gy, 40 Gy, 50 Gy, and 60 Gy isodose surface of the prescription dose, respectively. IMRT, intensity-modulated radiation therapy; IMPT, intensity-modulated proton therapy; NA, not applicable; SD, standard deviation.

## Data Availability

All data generated or analyzed during this study are included in this published article.

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
