# Peer review of "Salivary Gland Volume Changes and Dry Mouth Symptom Following Definitive Radiation Therapy in Oropharyngeal Cancer Patients—A Comparison of Two Different Approaches: Intensity-Modulated Radiation Therapy Versus Intensity-Modulated Radiation Therapy/Intensity-Modulated Proton Therapy Combination"

_cancers, 2025, doi:10.3390/cancers17030554_

Round 1

Reviewer 1 Report

Comments and Suggestions for Authors

This study compares salivary gland volume changes and dry mouth symptoms in 78 oropharyngeal cancer patients treated with IMRT alone or IMRT/IMPT combination. The results indicate that the critical factor in salivary gland volume change and dry mouth symptom was the delivered dose to the salivary glands, but not the RT technique used. This is an interesting study, however, there are some comments to consider, and the list of revision points is as follows:

1. The effects of RT on the sublingual glands were not evaluated in this study, this point needs to be addressed.

2. The authors must explain the criteria used to measure the dry mouth symptoms in the study to ensure clarity and reproducibility.

3. The authors need to add the Kappa value for intra-observer variability. Please check the term “inter-observer variability” in the discussion section.

4. In patients with comorbidities requiring medications that reduce salivary flow (medication-induced xerostomia and hyposalivation) should be excluded; this aspect should be stated in the discussion section.

5. The authors should address the clinical applications of the study findings in the discussion section such as relevance to clinical practice, application to specific patient populations, enhancement of treatment guidelines, and future directions.

6. The full term should accompany its abbreviation when it is first introduced in the text to provide clarity and help the reader understand it easily.

7. Please check the first paragraph in the results section. This paragraph should be removed from the manuscript.

8. The authors should revise the references to include those from the past 5-10 years, particularly in medical research, to ensure the information is up-to-date and reflects current knowledge and developments in the field.

Author Response

Comment 1: The effects of RT on the sublingual glands were not evaluated in this study, this point needs to be addressed.

Response: Thank you for highlighting this important point. The sublingual glands, while a critical component of the salivary gland system, were not included in this study due to their relatively small size and the difficulty in consistently delineating them on imaging studies. This limitation has now been explicitly addressed in the revised Materials and Methods section (Page 4, Line 149~152).

Comment 2: The authors must explain the criteria used to measure the dry mouth symptoms in the study to ensure clarity and reproducibility.

Response: Thank you for pointing this out. Dry mouth symptoms were evaluated retrospectively based on clinical assessments documented in the medical records. Severity was graded using the Common Terminology Criteria for Adverse Events (CTCAE) version 5.0, which classifies symptoms according to their impact on daily diet and the need for therapeutic intervention. This approach ensured consistency in the evaluation and allowed for reproducibility based on standardized criteria. We have included the criteria in the Materials and Methods section (Page 4, Line 143~146).

Comment 3: The authors need to add the Kappa value for intra-observer variability. Please check the term “inter-observer variability” in the discussion section.

Response: We appreciate your comment regarding the term "inter-observer variability". In our study, all contouring and volume measurements were carried out by a single radiation oncologist. This was specifically done to minimize inter-observer variability, ensuring that the data was consistent and reliable. We also acknowledge the importance of assessing intra-observer variability. However, due to the retrospective nature of the study and relatively small sample size, repeated measurements necessary to formally assess intra-observer variability were not feasible. Nonetheless, to reduce variability and ensure consistency in the delineation process, all salivary gland contours were delineated by a single radiation oncologist experienced in head and neck imaging.

Comment 4: In patients with comorbidities requiring medications that reduce salivary flow (medication-induced xerostomia and hyposalivation) should be excluded; this aspect should be stated in the discussion section.

Response: We appreciate the reviewer’s comment regarding the potential impact of medications that reduce salivary flow. In our study, patients with conditions or comorbidities requiring medications known to cause xerostomia or hyposalivation were excluded during the patient selection process. We will clarify this aspect in the Materials and Methods section (page 2, Line 84~88) to ensure transparency and address this concern.

Comment 5: The authors should address the clinical applications of the study findings in the discussion section such as relevance to clinical practice, application to specific patient populations, enhancement of treatment guidelines, and future directions.

Response: Thank you for your suggestion. We agree that highlighting the clinical applications of our findings is important. We have revised the discussion section (Page 10, Line 355~368) to address the relevance of our findings to clinical practice, their application to specific patient populations, potential contributions to enhancing treatment guidelines, and future research directions

Comment 6: The full term should accompany its abbreviation when it is first introduced in the text to provide clarity and help the reader understand it easily.

Response: Thank you for pointing this out. We have carefully reviewed the manuscript and ensured that all abbreviations are accompanied by their full terms upon their first introduction.

Comment 7: Please check the first paragraph in the results section. This paragraph should be removed from the manuscript.

Response: Thank you for identifying this issue.  We sincerely apologize for the oversight; the paragraph was mistakenly left in the manuscript during the editing process. We have now removed the paragraph from the results section as requested. We appreciate your careful review and understanding.

Comment 8: The authors should revise the references to include those from the past 5-10 years, particularly in medical research, to ensure the information is up-to-date and reflects current knowledge and developments in the field.

Response: Thank you for your valuable feedback. While we understand the importance of incorporating recent references to reflect current knowledge, as you may know, there has been a limited amount of research specifically addressing the salivary gland volume changes and dry mouth symptoms in the context of radiation therapy techniques such as IMRT and IMPT. Consequently, we had to rely on foundational and slightly older studies that are still highly relevant to our topic. Nevertheless, we have reviewed and updated the reference list to include more recent studies where possible. We appreciate your understanding and guidance in this matter.

Reviewer 2 Report

Comments and Suggestions for Authors

An interesting and good study. A topic which is highly relevant to this journal. The unique part of the study is the group using proton and photon combination which provide some additional insight on the effect of these mode of treatment to salivary glands.

The only thing that I would suggest to clarify is to add more details on how the xerostomia was assessed. I know it was mentioned this outcome was assessed by interview relating to the Common Terminology Criteria for Adverse Events version 5.0. But more detail on how the the question was done and the CTCAE is based on which toxicity ("Salivary duct inflammation"  I assume). Kindly add this. Thank you

Author Response

Comment 1: The only thing that I would suggest to clarify is to add more details on how the xerostomia was assessed. I know it was mentioned this outcome was assessed by interview relating to the Common Terminology Criteria for Adverse Events version 5.0. But more detail on how the the question was done and the CTCAE is based on which toxicity ("Salivary duct inflammation"  I assume).

Response: Thank you for your suggestion to provide additional details on how xerostomia was assessed. In our study, xerostomia was evaluated based on "dry mouth" criteria explicitly defined in the Common Terminology Criteria for Adverse Events version 5.0. The assessment relied on patient interviews during routine clinical visits, where symptoms were graded according to their impact on daily diet and the need for medical intervention. This information has been clarified in Materials and Methods section (Page 4, Line 143~146).

Reviewer 3 Report

Comments and Suggestions for Authors

I think the mss is well-written, but I also think no new information is provided. That xerostomia correlated with RT dose to the salivary glands is what is both expected and known, as you cite. Hence what new clinically relevant information are you reporting that would be of interest to readers?

That few add protons to IMRT narrows further the target audience.

(I also don’t fully understand your rationale for adding protons at all, but that is not the topic of your paper.)

Author Response

Comment: 

I think the mss is well-written, but I also think no new information is provided. That xerostomia correlated with RT dose to the salivary glands is what is both expected and known, as you cite. Hence what new clinically relevant information are you reporting that would be of interest to readers?

That few add protons to IMRT narrows further the target audience.

(I also don’t fully understand your rationale for adding protons at all, but that is not the topic of your paper.)

Response: 

We appreciate the reviewer’s insightful comments and understand the concerns raised. While it is well-established that xerostomia correlates with RT dose to the salivary glands, our study offers novel insights into the clinical application of combining IMRT with IMPT for treatment in a real-world setting.

Specifically, this study represents one of the first to investigate and compare the effects of IMRT alone versus an IMRT/IMPT combination on salivary gland volume changes and dry mouth symptoms over an extended follow-up period. By demonstrating no significant differences in outcomes between these approaches, despite the dosimetric advantages of IMPT in certain contexts, our findings provide a critical evaluation of how advanced RT techniques perform in practice, rather than in purely theoretical models or simulations.

Second, the study emphasizes the importance of RT dose to the salivary glands as the key determinant of xerostomia, irrespective of RT modality. This insight reinforces the need to optimize dosimetric planning to adhere to dose-volume constraints, even when advanced techniques like IMPT are employed. The findings also highlight the potential for tailored RT approaches to minimize side effects while maintaining oncologic efficacy, particularly in resource-limited settings where access to IMPT is restricted.

Third, the rationale for adding protons to IMRT in our clinical practice was to mitigate the long waiting times for IMPT alone, while still leveraging its dosimetric advantages. This approach enabled us to balance patient access with treatment efficacy. Although the rationale is not the central focus of the manuscript, the study provides valuable data on the outcomes of this combined approach, which may guide clinical decision-making in similar scenarios.

We believe our findings will contribute to the ongoing discussion about optimizing RT strategies for OPC patients and addressing the challenges of implementing proton therapy in routine clinical practice.

Round 2

Reviewer 1 Report

Comments and Suggestions for Authors

I would like to thank the authors for addressing my comments. The authors have sufficiently improved their paper, in reaction to the comments made.

Author Response

Thank you for the kind review.

Reviewer 3 Report

Comments and Suggestions for Authors

I appreciate your responses, but my concerns remain. There are reasons why IMPT is rarely used in this setting, something you do not discuss. Xerosomia is monitored regardless. 

Author Response

We appreciate the reviewer’s feedback and acknowledge the limited use of IMPT in this setting. The primary reason for its application in our study was to mitigate long waiting times while leveraging the dosimetric advantages of proton therapy. We recognize that IMPT is not a standard approach for all institutions, but our study provides real-world data on its feasibility in combination with IMRT. Additionally, while dry mouth is routinely monitored, our study specifically evaluates salivary gland volume changes over time and their correlation with dry mouth symptoms, offering further insight into treatment-related effects.